# TAD boundaries and gene activity are uncoupled

Faisal Almansour[1], Nadezda A Fursova[2], Adib Keikhosravi[3], Kathleen S Metz Reed[1], Daniel R Larson[2], Gianluca Pegoraro[3], Tom Misteli[1]*

[1]National Cancer Institute, NIH, Bethesda, United States; [2]Systems Biology of Gene Expression, National Cancer Institute, NIH, Bethesda, United States; [3]High-throughput Imaging Facility, National Cancer Institute, NIH, Bethesda, United States

## eLife Assessment

In this **important** study, DNA and RNA are co-imaged in single cells to show that the proximity of topologically associated domain (TAD) boundaries is uncoupled from the transcriptional activity of nearby genes. The evidence supporting these conclusions is **convincing** for the regions examined, with high-throughput imaging providing robust statistics. This work will be of interest to researchers studying genome architecture and its relationship to gene regulation.

**\*For correspondence:**
mistelit@mail.nih.gov

**Competing interest:** The authors declare that no competing interests exist.

**Abstract** Topologically associating domains (TADs) are prominent features of genome organization. A proposed function of TADs is to contribute to gene regulation by promoting chromatin interactions within a TAD and by suppressing interactions between TADs. Here, we directly probe the structure-function relationship of TADs by simultaneously assessing the behavior of TAD boundaries and gene activity at the single-cell and -allele level using high-throughput imaging. We find that while TAD boundaries pair more frequently than non-boundary regions, these interactions are infrequent and are uncorrelated with transcriptional activity of genes within the TAD. Similarly, acute global transcriptional inhibition or gene-specific activation does not alter TAD boundary proximity. Furthermore, while loss of the cohesin component RAD21 alters gene activity, disruption of TAD boundaries by depletion of the architectural chromatin protein CTCF is insufficient to alter expression of genes within the TAD. These results suggest that TAD boundary architecture and gene activity are largely uncoupled.

## Introduction

Beyond the linear DNA sequence, genomes are folded into higher-order structures (*Misteli, 2020*). Some of the most prominent genome features are chromatin loops and domains. Chromatin conformation mapping techniques—most notably Hi-C and Micro-C—have particularly highlighted topologically associating domains (TADs) as ubiquitous architectural features of higher eukaryotic genomes (*Dixon et al., 2012*; *Lieberman-Aiden et al., 2009*; *Nora et al., 2012*; *Sexton et al., 2012*).

TADs are self-assembling, contiguous genomic regions that preferentially interact with each other rather than with neighboring regions, creating distinct chromatin domains (*Lieberman-Aiden et al., 2009*). In human cells, TADs are typically 0.2–1 Mb in size and are defined by flanking boundary regions, which are marked by binding sites for the CTCF (CCCTC-binding factor) protein (*Lieberman-Aiden et al., 2009*; *Dekker and Mirny, 2016*; *Rowley M. and Corces V., 2018*). Mammalian TADs form via a process referred to as loop extrusion, which is driven by the association of the cohesin complex with DNA, and via its ATP-driven motor activity, reels in DNA until it encounters bound CTCF molecules at the TAD boundaries, thus forming a domain (*Dekker and Mirny, 2016*; *Rowley M. and*

*Corces V., 2018*; *Fudenberg et al., 2016*). In other organisms, chromatin domains form by similar mechanisms, although, for example, in *Drosophila*, only a small fraction of boundaries are CTCF-dependent (*Kaushal et al., 2021*).

TADs are thought to have a gene-regulatory function by bringing control elements, such as enhancers, over large genomic distances into physical proximity with their target genes within the same TAD, while at the same time limiting their interactions with genes in other TADs (*Dixon et al., 2012*; *Nora et al., 2012*). This model is in line with the known role of CTCF as an insulation factor (*Bell et al., 1999*; *Hark et al., 2000*). A regulatory role for TADs is also supported by the observation that targeted deletions or inversions of boundary elements alter enhancer-promoter communication and, in some cases, gene expression (*Chakraborty et al., 2025*; *Franke et al., 2016*; *Lupiáñez et al., 2015*). Furthermore, comparative mapping of the *Ubx* and *AbdA* TADs in *Drosophila* embryos showed that an ~2-fold change in enhancer-gene contact frequency leads to an ~7-fold difference in gene expression (*Mateo et al., 2019*). Disruption of TAD boundaries has also been linked to human disease, as structural variations that alter domain architecture can cause pathogenic rewiring of enhancer-gene contacts, e.g., in cancers, and in neurological and congenital disorders (*Spielmann et al., 2018*). Moreover, mutations in components of the cohesin complex cause developmental disorders, known as cohesinopathies, such as Cornelia de Lange syndrome, in which impaired chromatin architecture and altered transcriptional regulation are thought to drive the phenotype (*Dorsett and Krantz, 2009*).

On the other hand, several lines of evidence suggest that TADs are not strictly required for transcription regulation. Global removal of cohesin or CTCF produces surprisingly modest effects on genome-wide transcription, suggesting that most genes can be expressed relatively accurately without intact TAD structures (*Nora et al., 2017*; *Rao et al., 2017*). Although many CTCF and cohesin binding sites overlap with enhancers and promoters, a substantial fraction does not, indicating that domain boundaries are not universally tied to regulatory elements (*Kagey et al., 2010*; *Merkenschlager and Odom, 2013*; *Phillips-Cremins et al., 2013*). Furthermore, in *Drosophila*, large-scale rearrangements of chromatin domains lead to only modest transcriptional changes (*Ghavi-Helm et al., 2019*) and cis-regulatory transcription hubs form before domain establishment and prior to transcriptional activation, indicating that gene regulatory contacts may emerge independently of domain architecture (*Espinola et al., 2021*). Similarly, during dorsoventral patterning in *Drosophila*, tissue-specific gene expression patterns emerge despite largely invariant chromatin domains across tissues (*Ing-Simmons et al., 2021*). Finally, in mammals, enhancer-promoter contacts persist even after the global loss of CTCF or cohesin, underscoring that regulatory interactions can be maintained without stable TAD anchoring (*Chakraborty et al., 2023*; *Hsieh et al., 2022*; *Platania et al., 2024*; *Taylor et al., 2022*). These observations point to a limited functional role of TAD architecture in gene regulation.

A confounding factor in assessing the functional role of TADs on gene regulation is the recent realization that TAD structure is highly dynamic, resulting in variable TAD conformations in individual cells and alleles (*Bintu et al., 2018*; *Cattoni et al., 2017*; *Finn et al., 2019*; *Gabriele et al., 2022*). High-throughput DNA FISH studies indicate that TAD boundary pairing only occurs in typically 5–20% of alleles at any given time in a population (*Finn et al., 2019*). In agreement, live-cell imaging demonstrates that TAD boundaries undergo continuous motion, and that the persistence time of pairing is on the order of ~10–30 min before they separate again, consistent with polymer simulations of cohesin-mediated loops (*Gabriele et al., 2022*; *Sabaté et al., 2023*). In addition to the variable nature of TAD architecture, gene expression itself is similarly dynamic, with most genes undergoing rapid cycles of activity and inactivity, referred to as gene bursting (*Rodriguez et al., 2019*; *Sood et al., 2025*). Furthermore, recent observations point to a bidirectional relationship in which transcriptional activity itself also affects chromatin structure (*Platania et al., 2024*; *Chahar et al., 2023*; *Luppino et al., 2022*; *Shaban et al., 2024*). Neither the variability in TAD organization nor the dynamics of gene activity is captured by commonly used population-based profiling methods, confounding the assessment of the effect of TAD structure on gene expression at the level of individual alleles.

Here, we directly probe the structure-function relationship of TADs at the single-cell and single-allele level by use of high-throughput imaging to simultaneously visualize TAD boundaries by DNA-FISH and nascent RNA production by RNA-FISH. Using the TADs containing the *EGFR* and *MYC* genes, respectively, as model systems, we quantitatively compare boundary distances at individual active and inactive alleles or upon transcriptional perturbation or stimulation. We also probe the

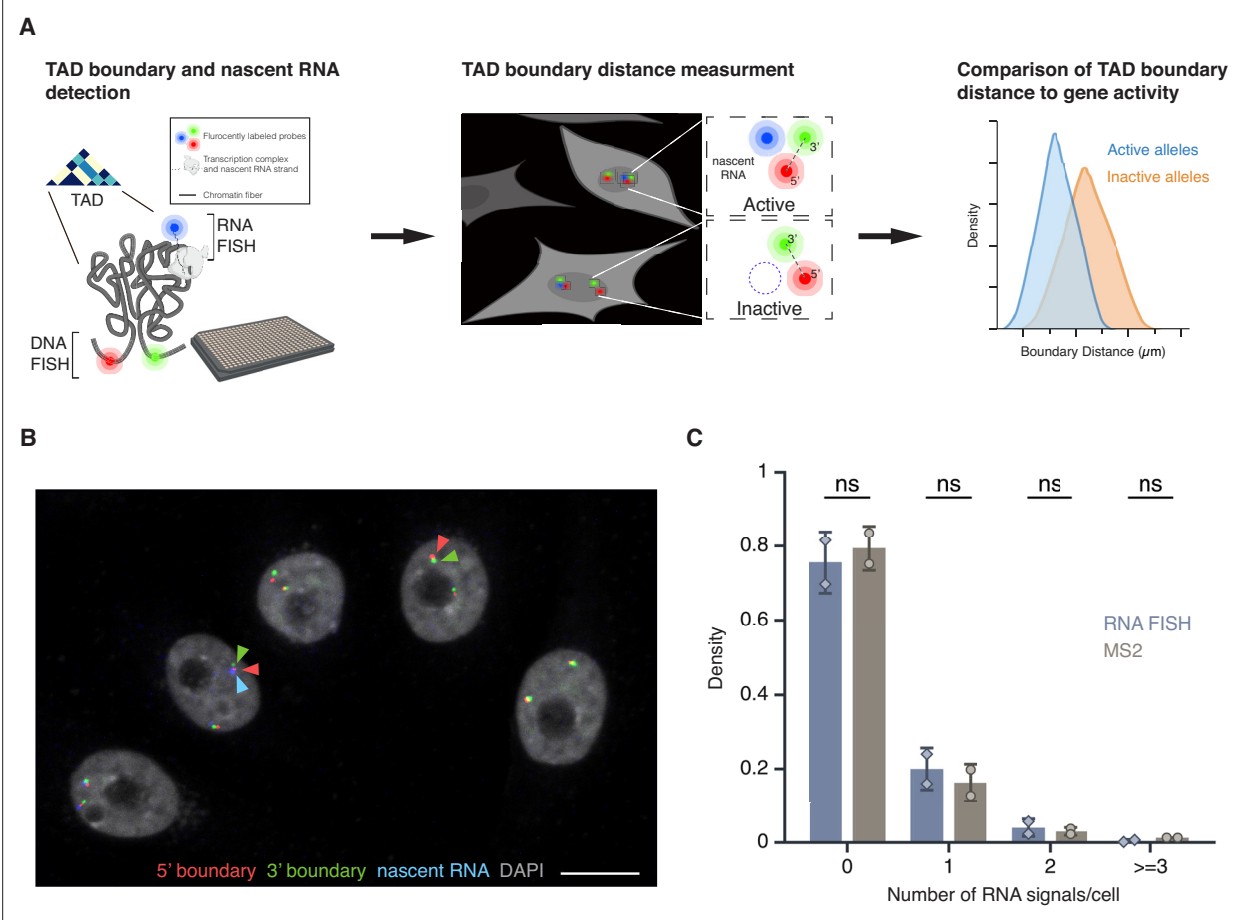

**Figure 1.** High-throughput DNA/RNA FISH. (**A**) Schematic overview of the DNA/RNA high-throughput FISH imaging (HiFISH) pipeline used to simultaneously measure topologically associating domain (TAD) boundary distance and gene activity at the single-cell and single-allele levels. Step 1: Design of DNA FISH probes based on Micro-C profiling and detection of DNA and nascent RNA by HiFISH. Step 2: Measurement of center-to-center TAD boundary distances and RNA signal at individual alleles by image analysis using HiTIPS (*Keikhosravi et al., 2024*). Step 3: Quantitative comparison of TAD boundary distances with gene activity at each allele. (**B**) DNA/RNA HiFISH detection of 5' (green) and 3' (red) *MYC* TAD boundaries and nascent RNA (blue) in human bronchial epithelial cells (HBECs). Scale bar: 10 µm. (**C**) Quantification of *MYC* nascent RNA signals using DNA/RNA HiFISH in fixed HBECs or an MS2-tagged *MYC* reporter in living HBECs. Bars represent means ± SEM from two experiments. Dots indicate means from individual experiments. 166,953 cells were analyzed for MS2, and 30,137 cells for DNA/RNA HiFISH. Statistical significance was calculated using two-way ANOVA with Bonferroni correction: ns, not significant (p≥0.05).

effect of loss of the architectural TAD protein CTCF on gene expression. We find that TAD boundary proximity is unrelated to gene activity.

## Results

### Simultaneous assessment of TAD boundaries and gene activity by high-throughput imaging

To quantitatively analyze the relationship between TAD boundaries and gene activity at the single-cell and single-allele level, we developed a high-throughput FISH imaging (HiFISH) and image analysis pipeline (*Almansour et al., 2024*; *Keikhosravi et al., 2024*) comprised of three components: (1) detection of TAD boundaries and nascent RNA using combined DNA- and RNA-FISH in a 384-well high-throughput format (DNA/RNA HiFISH) (*Almansour et al., 2024*), (2) measurement of center-to-center TAD boundary distances using HiTIPS, a customized image analysis software to probe features of nuclear architecture (*Keikhosravi et al., 2024*), and (3) quantitative comparison of boundary distances

and gene expression status at each visualized allele (*Figure 1A*; see Materials and Methods). DNA/RNA HiFISH was performed simultaneously in a single hybridization step, as previously described (*Almansour et al., 2024*). Boundary distances were measured in 2D maximum-intensity projections generated from 3D imaging stacks using center-to-center distance measurements and a pixel resolution of 152 nm, as previously described (*Finn et al., 2017*; see Materials and Methods). Similar results were obtained using 2D maximum-intensity projections and 3D imaging (*Finn et al., 2017*; see below).

The combined DNA/RNA HiFISH imaging approach resulted in robust simultaneous detection of TAD boundaries and nascent RNA in multiple cell types (*Almansour et al., 2024*; *Figure 1B*; see below). The correct number of DNA FISH signals was routinely detected in >95% of cells in non-transformed hTERT-HFFc6 fibroblasts (HFFs) or in human bronchial epithelial cells (HBECs), as previously reported (*Almansour et al., 2024*). Similarly, the detected RNA-FISH signals accurately reflected the number of active alleles as demonstrated by comparison with the number of active *MYC* alleles visualized by live-cell imaging using an MS2-tagged *MYC* reporter system in HBECs (*Figure 1C*). The high detection efficiency underscores the sensitivity and specificity of our approach for probing TAD boundary distances and nascent transcription at the single-cell and single-allele level (*Almansour et al., 2024*). The high-throughput nature of this approach enabled routine probing of thousands of individual alleles per experimental condition, providing high statistical power in comparative analyses.

## Selection and validation of model TADs

Two TADs containing the *EGFR* and *MYC* genes, respectively, were selected as models for our analysis based on their high biological relevance in signaling and transcription, respectively, and their presence in structurally well-defined TADs, as mapped by high-resolution (1 kb) publicly available Micro-C datasets of HFF cells (*Krietenstein et al., 2020*) and human embryonic stem cells (ESCs) (*Akgol Oksuz et al., 2021*). Both TADs are conserved in both cell types and display well-defined corner peaks and side streaks, both hallmarks of stable and structurally distinct TADs (*Krietenstein et al., 2020*; *Akgol Oksuz et al., 2021*).

The *EGFR* TAD spans ~500 kb and has two sub-TADs of ~250 kb each, while the *MYC* TAD extends over ~3 Mb and comprises two large sub-TADs (~1 Mb and ~2 Mb, respectively) (*Figure 2A*). The *MYC* TAD is more insulated from flanking chromatin than the *EGFR* TAD, which itself may reside in a sub-TAD within a broader domain not visible at lower Hi-C resolution (*Krietenstein et al., 2020*; *Akgol Oksuz et al., 2021*). The two TADs also differ in their gene content and chromatin landscape based on ChromHMM analysis (*Ernst and Kellis, 2012*; *Figure 2—figure supplement 1*), with the *EGFR* TAD enriched in transcriptionally active regions, while the *MYC* TAD is predominantly quiescent, with isolated active chromatin features clustered near the long noncoding RNAs *PVT1* and *PCAT1*. The *MYC* TAD contains only two protein-coding genes, *MYC* and *POU5F1B,* and multiple noncoding elements (*Figure 2—figure supplement 1*). The *MYC* gene lies close to the 5′ boundary, while *POU5F1B* is within the upstream sub-TAD. The *EGFR* TAD harbors three protein-coding genes—*EGFR*, *LANCL2*, and *VOPP1*—with the *EGFR* gene positioned relatively distally upstream (*Figure 2—figure supplement 1*). The fraction of the TAD covered by each gene also varies: *EGFR* occupies ~40% of its TAD, whereas *LANCL2* and *VOPP1* cover ~17% and ~20%, respectively. In contrast, *MYC* and *POU5F1B* together span less than 1% of the *MYC* TAD (*Figure 2—figure supplement 1*). The *MYC* and *EGFR* TAD boundaries lie in largely quiescent chromatin, with the exception of the 3′ *EGFR* TAD boundary, which contains some active marks due to the proximity to the *EGFR* gene (*Figure 2—figure supplement 1*). The differences in size and variation in structural and functional features between the two TADs make them attractive and robust models for investigating the relationship between TAD boundaries and gene activity at single-allele resolution.

To detect the *EGFR* and *MYC* TAD boundaries, we selected specific BAC DNA FISH probes of typically ~165 kb that directly target the 5′ and 3′ boundaries, respectively (*Figure 2A*; *Figure 2—figure supplements 1 and 2*; *Supplementary file 1*). The large probe size ensures high detection sensitivity without loss of accuracy as previously described (*Finn and Misteli, 2021*). To control for regional variability and to distinguish boundary-specific behavior from broader chromatin effects, control DNA FISH probes targeting non-TAD control regions were also used. These probes were positioned on the same chromosome arm, equidistant upstream of the respective 5′ TAD boundaries and located within the same or adjacent cytogenetic bands (*Figure 2A*; *Figure 2—figure supplements 1 and 2*). The *EGFR*-associated non-TAD control region contains one protein-coding gene and fewer noncoding

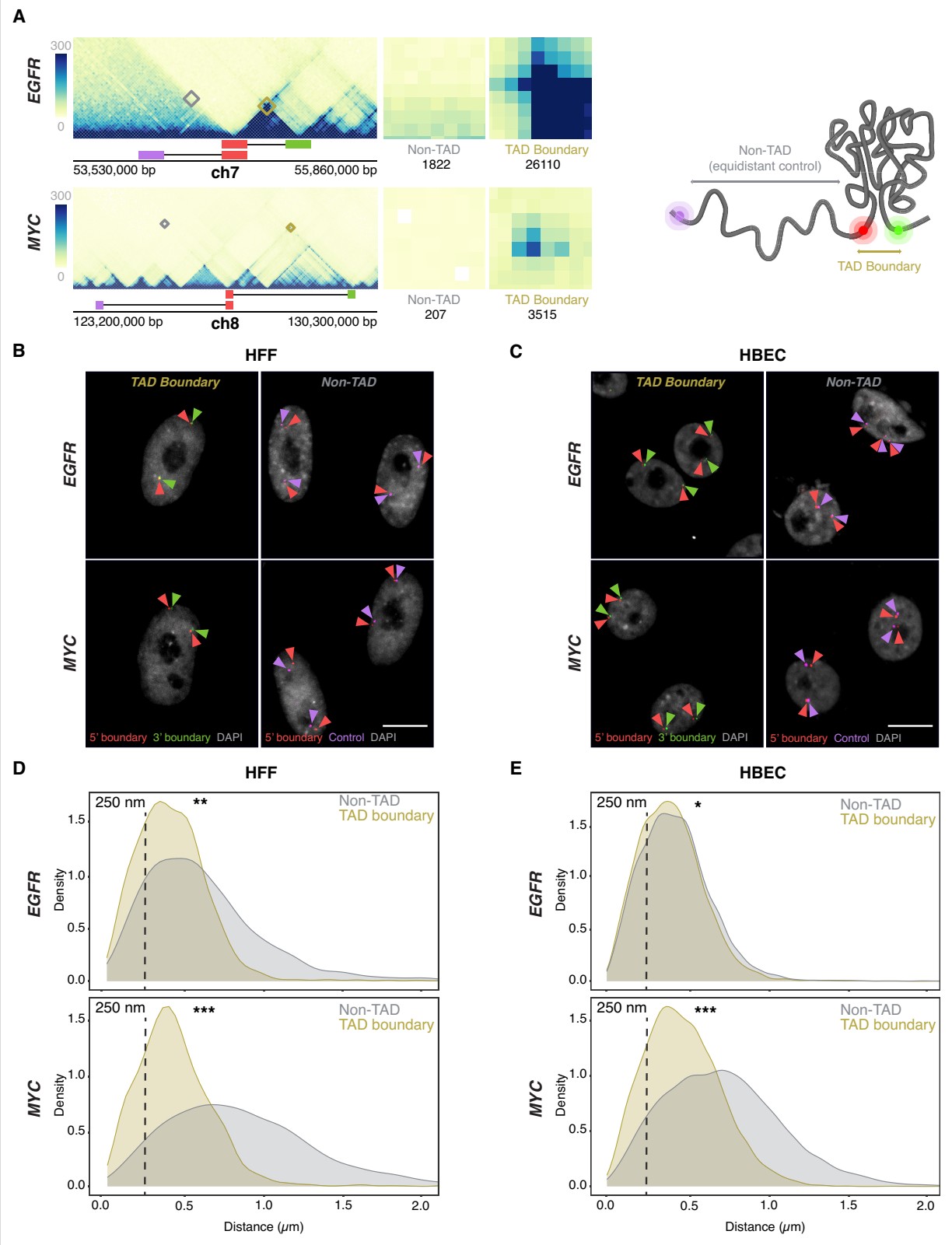

**Figure 2.** Topologically associating domain (TAD) boundaries interact more frequently than non-TAD regions. (**A**) Micro-C contact maps for *EGFR* and *MYC* TADs and adjacent regions in hTERT-HFFc6 fibroblast (HFF) cells, highlighting TAD boundaries and genomically equidistant non-TAD regions. Squares denote the probe positions used for 3' (green), 5' boundary (red), and equidistant non-TAD controls (purple). Interactions between the 5' TAD boundaries and the 3' TAD (yellow) or non-TAD boundaries (gray) are highlighted, and total Micro-C contacts between regions are

*Figure 2 continued on next page*

*Figure 2 continued*

quantified, emphasizing high TAD boundary contact frequency in both *EGFR* and *MYC* TADs, as well as weaker signals in the non-TAD regions. (**B–C**) Representative DNA high-throughput FISH imaging (HiFISH) images of *EGFR* and *MYC* TAD boundary and non-TAD regions in HFF cells (**B**) and human bronchial epithelial cells (HBECs) (**C**). Scale bar: 10 μm. (**D–E**) Measurement of boundary distances. Distance distributions of *EGFR* and *MYC* TAD boundaries vs. matched non-TAD regions in HFF cells (**D**) and HBECs (**E**). Dashed line indicates 250 nm threshold used to define physical interaction. Between 2,000 and 18,000 alleles were analyzed per sample. Values represent an individual dataset from a single experiment of multiple experiments. Mann-Whitney U test p-values are: ***$p < 1 \times 10^{-100}$; ** $1 \times 10^{-100} \le p < 1 \times 10^{-20}$; * $1 \times 10^{-20} \le p < 0.01$.

The online version of this article includes the following figure supplement(s) for figure 2:

**Figure supplement 1.** Micro-C chromosome interaction maps and ChromHMM analysis of *EGFR, MYC, ERRFI1, FKBP5,* and *VARS2* topologically associating domains (TADs) in HFFc6.

**Figure supplement 2.** Sequence and location of DNA and RNA probes binding sites for DNA/RNA high-throughput FISH imaging (HiFISH).

elements relative to its corresponding TAD boundary (*Figure 2—figure supplement 1*). In contrast, the *MYC* non-TAD control region contains multiple genes and exhibits a more complex local topology, bordering multiple looped domains and small TADs (*Figure 2—figure supplement 1*). Both non-TAD regions share a relatively quiescent chromatin state, with sparse transcriptional and regulatory element enrichment. Consistent with their function as boundaries, Micro-C contact frequencies were at least 10-fold higher for the *MYC* and *EGFR* TAD boundary regions compared to the corresponding control regions (*Figure 2A*).

## TAD boundaries pair at low frequency

We first used DNA/RNA HiFISH to determine the distance distribution profiles and interaction frequencies of TAD boundaries at the single-allele level (*Almansour et al., 2024*; *Figure 2B–E*; see Materials and Methods). As expected, the median distance for *MYC* TAD boundaries was smaller compared to non-TAD regions in both HFF (0.41±0.30 μm vs. 0.81±0.63 μm; median ±interquartile range) and HBEC (0.46±0.33 μm vs. 0.71±0.50 μm; Mann-Whitney U test, $p < 1 \times 10^{-10}$ for both) (*Figure 2D and E*). Similarly, in HFF, *EGFR* TAD boundaries were in closer proximity than non-TAD regions (0.41±0.30 μm vs. 0.56±0.48 μm; U test, $p < 1 \times 10^{-20}$) (*Figure 2D*). Interestingly, the *EGFR* TAD boundaries exhibited a more uniform distribution in HBEC, with TAD boundaries showing a similar distribution to non-TAD regions (0.39±0.28 μm vs. 0.42±0.29 μm; U test, $p < 0.01$) (*Figure 2E*). The lack of a strong difference in proximity at *EGFR* boundaries in HBECs likely reflects the smaller size of the *EGFR* TAD (~0.5 Mb) (*Figure 2A*; *Figure 2—figure supplement 1*) and the smaller nuclear size in HBECs (*Figure 2C*).

To quantify the pairing frequency of *MYC* and *EGFR* TAD boundaries, we calculated the percentage of alleles with TAD boundary distances below a 250 nm threshold, a value previously used to define chromatin interactions (*Finn et al., 2019*; *Gabriele et al., 2022*). *EGFR* TAD boundaries were within 250 nm in 31 ± 14% (mean ± SD) of alleles in HBEC and 33 ± 17% in HFF, and *MYC* TAD boundaries were within this range in 23 ± 9% of HBEC and 27 ± 9% of HFF alleles (*Figure 2D and E*). Non-TAD control regions showed lower interaction frequencies with 27 ± 13% and 20 ± 8% for *EGFR* in HBEC and HFF, respectively, and 8 ± 1% and 6 ± 0.3% for *MYC* in HBEC and HFF (*Figure 2D and E*; $p < 0.05$ for all comparisons). As previously noted, while adjusting the distance threshold changes the absolute percentage of close contacts, it does not affect the relative differences between TAD boundaries and non-TAD regions (*Finn et al., 2019*). These results are consistent with single-cell FISH and Hi-C studies showing that TAD boundaries pair two- to threefold more frequently than non-TAD regions, and with live-cell imaging studies showing transient boundary pairing (*Cattoni et al., 2017*; *Finn et al., 2019*; *Gabriele et al., 2022*; *Carstens et al., 2016*; *Flyamer et al., 2017*; *Giorgetti et al., 2014*; *Nagano et al., 2013*; *Stevens et al., 2017*; *Szabo et al., 2018*). Our findings confirm that TAD boundaries exhibit higher interaction frequencies and shorter distances than non-TAD regions, but that boundary pairing is a relatively infrequent and transient event, as previously observed by FISH (*Finn et al., 2019*) and live-cell imaging (*Gabriele et al., 2022*).

## TAD boundary distance is not related to gene activity status

To assess whether TAD boundary proximity correlates with gene activity at the single-allele level, boundary distances were compared between transcriptionally active and inactive alleles using

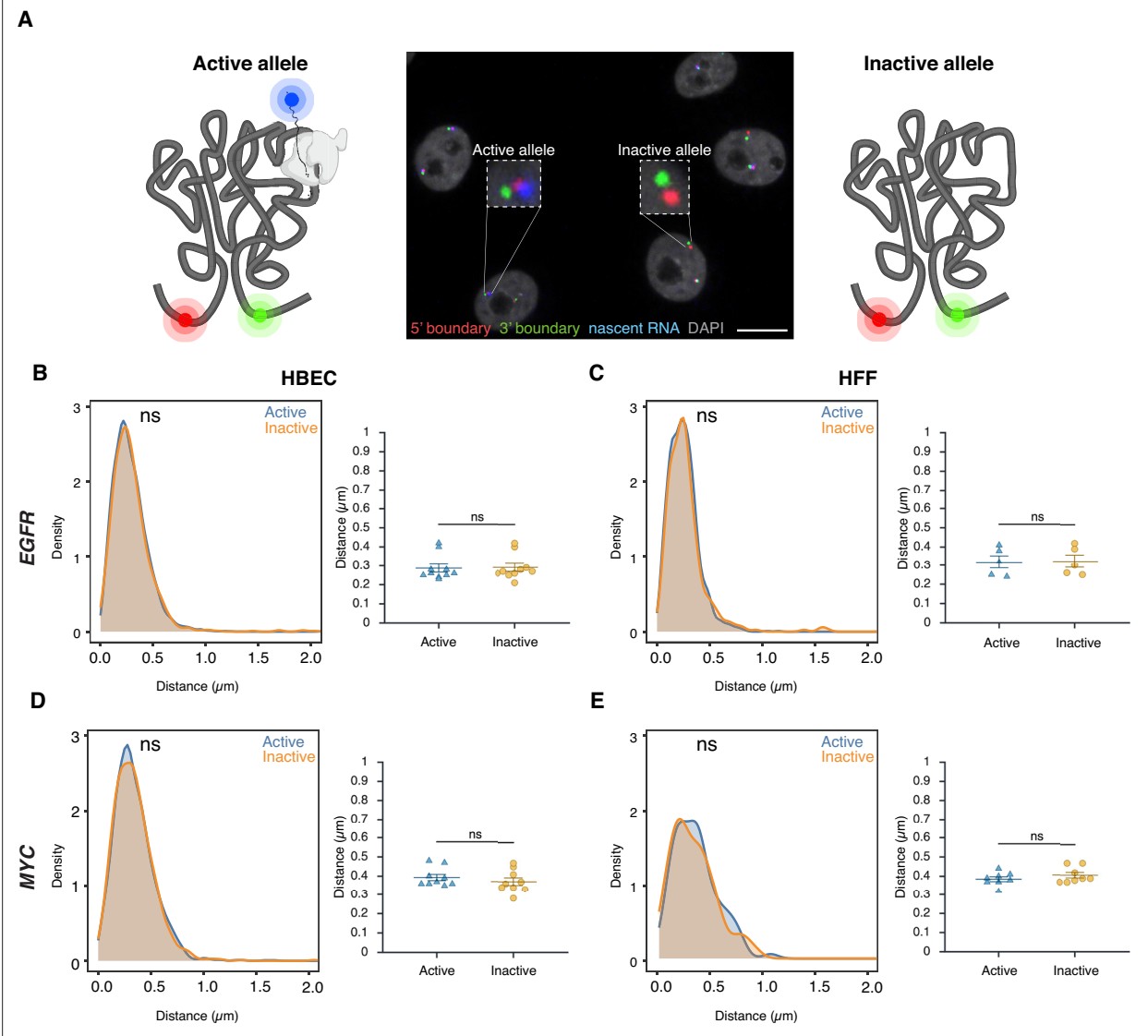

**Figure 3.** Topologically associating domain (TAD) boundary proximity is not related to gene activity status. (**A**) Representative DNA/RNA high-throughput FISH imaging (HiFISH) image of *EGFR* nascent RNA (blue) and its associated 5' (red) and 3' (green) TAD boundaries in HBECs, illustrating detection of active (RNA-positive) and inactive (RNA-negative) alleles. Scale bar: 10 µm. (**B – E**) Comparison of TAD boundary distances for *EGFR* and *MYC* alleles based on transcriptional activity status. Histograms of allele-specific distance distributions from a representative dataset from a single experiment; Mann-Whitney U test p-values are indicated as follows: ns, not significant (p≥0.05). Dot plots of the mean of median distances from multiple experiments (500–20,000 alleles per condition); error bars represent SEM, and statistical significance was calculated using two-way ANOVA with Bonferroni correction: ns, not significant (p≥0.05).

© 2026, Guin. Panel A was created with BioRender and is published under a Creative Commons Attribution License. Further reproductions must adhere to the terms of this license.https://creativecommons.org/licenses/by/4.0/

The online version of this article includes the following figure supplement(s) for figure 3:

**Figure supplement 1.** Topologically associating domain (TAD) boundary proximity is uncoupled from allelic gene activity in single nuclei in both 2D and 3D imaging.

high-throughput DNA/RNA HiFISH (*Figure 3A*). Active alleles were identified by the presence of nascent RNA FISH signals in the proximity (<1 µm) of a TAD boundary signal (see Materials and Methods), while inactive alleles lacked detectable RNA signals near TAD boundary signals (*Figures 1A and 3A*). RNA detection was efficient, as indicated by the comparable number of active transcription sites detected by RNA-FISH as in living cells using the MS2-RNA detection system (*Figure 1C*).

Comparative analyses of multiple independent DNA/RNA HiFISH experiments—each comprising up to 20,000 alleles—revealed no consistent difference in TAD boundary distances between active and inactive alleles of *EGFR* and *MYC* loci in HBEC, HFF, and HCT116 cells (*Figure 3B–E*; *Figure 3—figure supplement 1*). In HBEC, *EGFR* boundary distances were identical between active (0.29±0.07 µm, mean of medians ± SD) and inactive (0.29±0.07 µm) alleles (Mann-Whitney U test, p=0.57) (*Figure 3B*). Similarly, in HFF, both active and inactive *EGFR* alleles exhibited identical boundary distances (0.32±0.08 µm; p=0.75) (*Figure 3C*). A similar pattern was observed for *MYC*. In HBEC, median boundary distances were 0.39±0.05 µm for active *MYC* alleles and 0.37±0.06 µm for inactive alleles (p=0.31) (*Figure 3D*). In HFF, active and inactive *MYC* alleles also had similar distances (0.39±0.04 µm vs. 0.41±0.04 µm; p=0.46) (*Figure 3E*). A similar pattern was observed for both genes in HCT116 (*Figure 3—figure supplement 1*). Furthermore, the TAD boundary distances of the two alleles in the same nucleus were uncorrelated (*Figure 3—figure supplement 1*) and not significantly different between active and inactive alleles, either as measured by 2D or 3D imaging (*Figure 3—figure supplement 1*). Similar results were observed when the data were stratified by distance rather

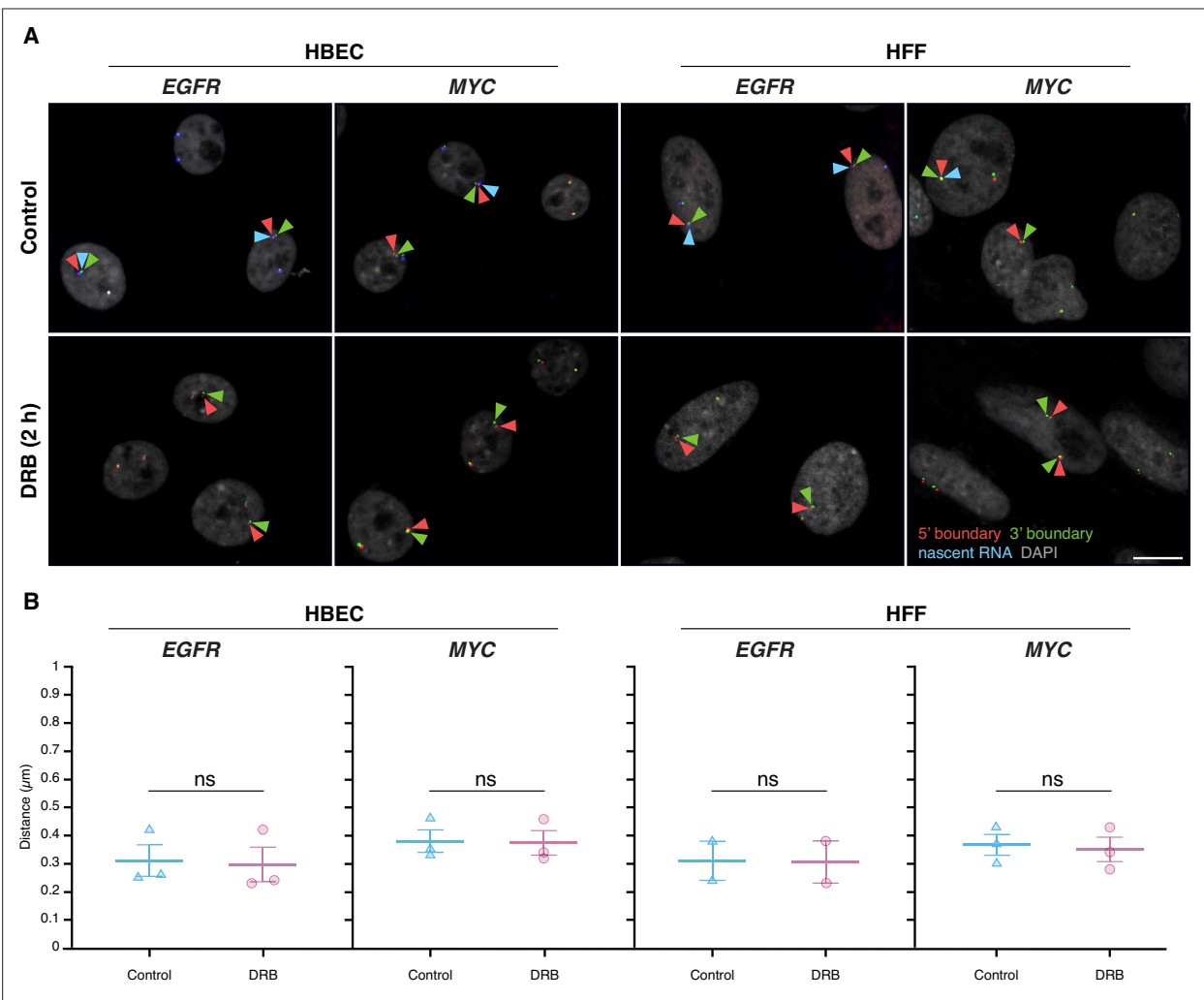

**Figure 4.** Global transcription inhibition does not alter topologically associating domain (TAD) boundary pairing. (**A**) Representative DNA/RNA high-throughput FISH imaging (HiFISH) images of *EGFR* and *MYC* TAD boundary and nascent RNA in HBECs and HFF cells with and without 2 hr 5,6-dichlorobenzimidazole 1-β-D-ribofuranoside (DRB) treatment. Scale bars: 10 µm. (**B**) Quantification of TAD boundary distances for *EGFR* and *MYC* in the presence or absence of DRB. Dot plots of the mean of median distances from multiple experiments (500–20,000 alleles per condition). Error bars represent SEM. Statistical significance was calculated using two-way ANOVA with Bonferroni correction: ns, not significant (p≥0.05).

The online version of this article includes the following figure supplement(s) for figure 4:

**Figure supplement 1.** Transcriptional inhibition does not affect the spatial organization of non-topologically associating domain (TAD) control regions.

than activity status (*Figure 3—figure supplement 1*). Taken together, these results indicate that the proximity of *MYC* and *EGFR* TAD boundaries is not related to gene activity at individual alleles.

## Inhibition of gene activity does not alter TAD boundary pairing

To further test the relationship between TAD boundaries and gene activity, we assessed whether acute global transcriptional inhibition alters TAD boundary proximity. We treated HBEC or HFF cells with 5,6-dichlorobenzimidazole 1-β-D-ribofuranoside (DRB), an inhibitor of CDK9 and CDK7 that acutely blocks RNA polymerase II (RNAPII) transcription elongation and initiation (*Baumli et al., 2010*; *Rahl et al., 2010*). As expected, after 2 hr of DRB treatment, nascent RNA signals for *EGFR* and *MYC* decreased by over 90%, confirming effective transcriptional inhibition (*Figure 4A*).

TAD boundary distances remained unchanged for both *EGFR* and *MYC* upon transcriptional inhibition (*Figure 4B*). In HBEC and HFF cells, *EGFR* TAD boundary median distances were similar in untreated controls (0.31±0.10 µm and 0.31±0.10 µm; mean of medians ± SD) and DRB-treated cells (0.30±0.11 µm and 0.31±0.11 µm; U-test p-values = 0.51 and 1.00, respectively). Likewise, *MYC* TAD boundary distances showed minimal changes in HBEC (0.38±0.07 µm control vs. 0.37±0.08 µm treated; p=0.83) nor in HFF (0.37±0.07 µm control vs. 0.35±0.08 µm treated; p=0.83) (*Figure 4B*). Transcriptional inhibition also did not affect the distance distribution of non-TAD control regions

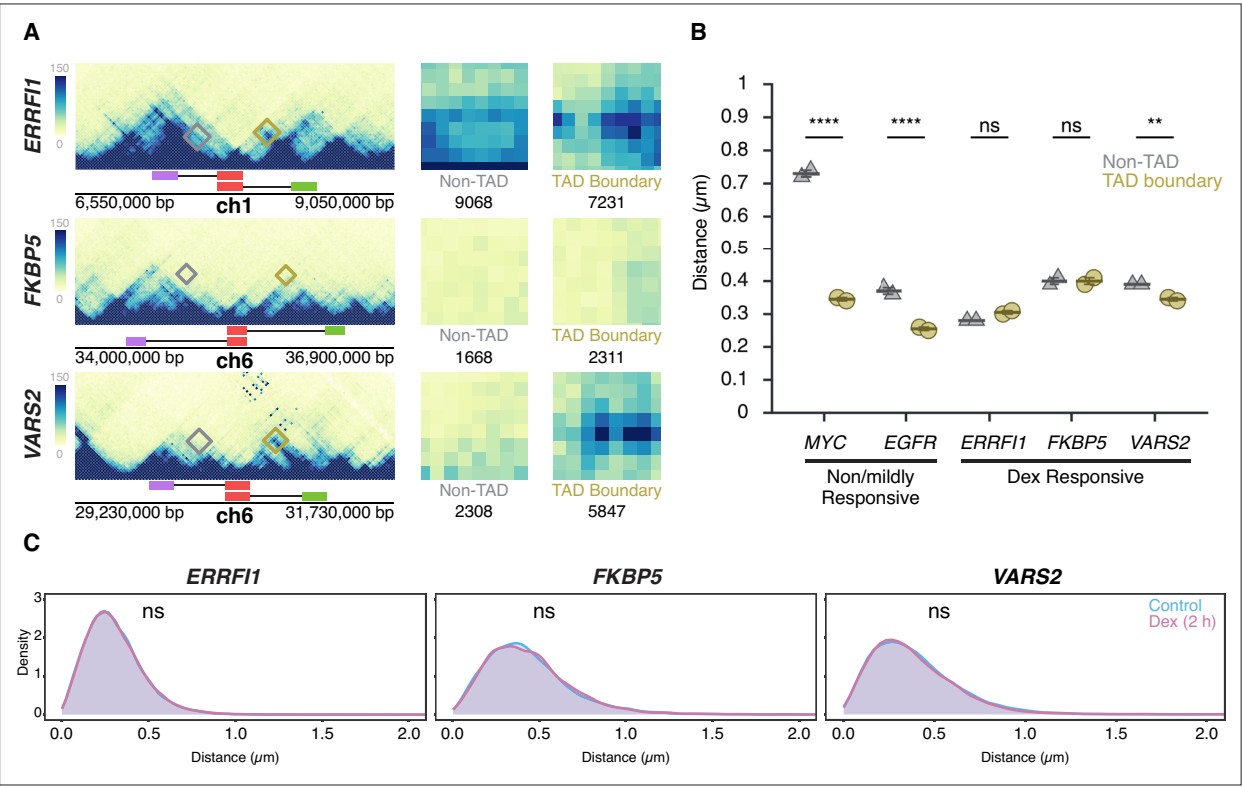

**Figure 5.** Transcription stimulation does not alter topologically associating domain (TAD) boundary interactions. (**A**) Micro-C maps for the *ERRFI1*, *FKBP5*, and *VARS2* TADs and neighboring regions, showing TAD boundaries (green, red) and equidistant non-TAD control regions (red, purple) in hTERT-HFFc6 fibroblast (HFF); corresponding probe positions are indicated. Interactions between the 5' TAD boundaries and the 3' TAD (yellow) or equidistant non-TAD control regions (gray) are highlighted, and total Micro-C contacts between regions are quantified, showing prominent contact frequency between *ERRFI1*, *VARS2*, and *FKBP5* TAD boundaries, as well as the non-TAD region of *ERRFI1*. (**B**) Comparison of TAD boundary and non-TAD region distances for *EGFR*, *MYC*, *ERRFI1*, *FKBP5*, and *VARS2* in human bronchial epithelial cells (HBECs) as measured by DNA HiFISH. Dot plots of the mean of median distances from two experiments (11,000–49,000 alleles per condition). Error bars represent SEM. Statistical significance was calculated using two-way ANOVA with Bonferroni correction: ****p<0.0001; **p<0.01; ns, not significant (p≥0.05). (**C**) Measurement of boundary distances. Distance distributions of *ERRFI1*, *FKBP5*, and *VARS2* TADs in untreated and 2 hr dexamethasone (Dex)-treated HBEC. Between 2,500 and 6,000 alleles were analyzed per condition. Values represent an individual dataset from a single experiment representative of multiple experiments. Mann-Whitney U test p-values are indicated as follows: ns, not significant (p≥0.05).

The online version of this article includes the following figure supplement(s) for figure 5:

**Figure supplement 1.** RNA levels following dexamethasone (Dex) treatment.

(*Figure 4—figure supplement 1*). These results indicate that acute inhibition of global RNAPII-dependent transcription does not significantly impact TAD boundary distances, suggesting that the behavior of TAD boundaries is uncoupled from short-term gene expression dynamics.

## Stimulation of gene activity does not change TAD boundary distances

To conversely assess whether transcriptional activation influences TAD boundaries, HBECs were treated for 2 hr with dexamethasone (Dex), a glucocorticoid receptor (GR) agonist known to selectively induce GR-target genes (*Bothe et al., 2021*). We selected for this analysis *ERRFI1*, *FKBP5*, and *VARS2*, which are all robustly induced (2- to 7- fold) as measured by RNA-seq (*Figure 5—figure supplement 1*) and reside in relatively large TADs of 400, 1000, and 700 kb, respectively (*Figure 5A*). Interestingly, unlike in TADs for the Dex-insensitive *MYC* and *EGFR* genes, for the inducible *ERRFI1*, *FKBP5*, and *VARS2* genes TAD boundary distances were largely similar to the corresponding non-TAD boundry controls (*Figure 5B*). Distances for TAD boundaries were 0.31±0.01 μm (mean of medians ± SD) 0.40±0.02 μm, and 0.35±0.01 μm for *ERRFI1*, *FKBP5*, and *VARS2*, respectively, and were comparable to those for non-TAD regions (0.28±0.00 μm, 0.40±0.01 μm, and 0.39±0.00 μm). Statistical comparison by Bonferroni's multiple-comparison test of TAD vs. non-TAD distances showed no significant difference for *ERRFI1* (p=0.1572, ns) or *FKBP5* (p=1.0000, ns), but a modest yet significant difference for *VARS2* (p=0.0057) (*Figure 5B*).

As expected, based on the increased steady-state RNA levels detected by RNA-seq data upon Dex stimulation, the number of cells with one or two nascent RNA signals for *ERRFI1*, *FKBP5*, and *VARS2* increased after Dex treatment for either 2 or 4 hr compared to untreated controls (*Figure 5—figure supplement 1*). However, despite robust transcriptional induction, TAD boundary distances remained unchanged (p>0.3 for all comparisons) (*Figure 5C*). The proportion of alleles within 250 nm also did not differ between Dex-treated and control conditions (Mann-Whitney U test, p>0.2) (*Figure 5C*). Together, these results demonstrate that TAD boundary proximity is unaffected by acute transcriptional activation, reinforcing the notion that TAD boundary structure and gene activity are uncoupled.

## TAD boundary architecture and gene expression

Our results indicate that gene expression status does not affect TAD boundaries. We next asked whether, conversely, alterations in TAD boundary structure affect transcription. To do so, we depleted the cohesin complex component RAD21 or the boundary protein CTCF in HCT116 cells using previously characterized auxin-inducible degron (AID) systems (*Rao et al., 2017*; *Natsume et al., 2016*; *Yesbolatova et al., 2020*) and assessed the effect of depletion of either factor on boundary structure and transcription by DNA/RNA HiFISH (*Figure 6*; *Figure 6—figure supplement 1*).

Consistent with prior studies, treatment of HCT116-RAD21-AID cells with auxin for 3 or 6 hr resulted in near-complete degradation of RAD21 (*Rao et al., 2017*; *Natsume et al., 2016*; *Figure 6—figure supplement 1*). RAD21 depletion significantly increased the 5'–3' TAD boundary distances for both the *EGFR* and *MYC* TADs (*Figure 6A and B*). For *EGFR*, the median boundary distance increased from 0.25±0.20 μm (control; median ±IQR) to 0.39±0.36 μm (RAD21-depleted, Mann-Whitney U test, p<1 × 10$^{-10}$). For *MYC*, the median increased from 0.34±0.26 μm to 0.49±0.49 μm (U test, p<1 × 10$^{-10}$). The fraction of alleles with TAD boundaries within 250 nm decreased from 49% to 26% for *EGFR* and from 29% to 18% for *MYC* upon depletion of RAD21 (*Figure 6B*). RAD21 depletion also reduced the expression of *EGFR* and *MYC*, with the median number of transcription sites per cell decreasing by 1.6-fold and 2.1-fold, respectively (*Figure 6C*). In line with gene repression, the percentage of cells with no detectable nascent RNA signal increased for both *EGFR* and *MYC* (Bonferroni-adjusted p-values = 2.70e-3 and <0.0001, respectively) (*Figure 6C*), and monoallelic expression frequencies also modestly decreased (*Figure 6C*). Similar results were observed for the *ERRFI1 gene* upon loss of RAD21 (*Figure 6—figure supplement 1*).

The reduction in gene expression upon loss of RAD21 may either be due to changes in TAD architecture or, more likely, due to local effects of RAD21, such as in enhancer-promoter interactions (*Kagey et al., 2010*; *Merkenschlager and Odom, 2013*; *Phillips-Cremins et al., 2013*). To more directly assess a possible role of TAD structure on gene expression, we depleted the boundary factor CTCF via degron as previously described (*Yesbolatova et al., 2020*; *Figure 6—figure supplement 1*). Depletion of CTCF increased *MYC* TAD boundary distances (median increased from 0.33±0.25 μm to 0.44±0.30 μm; Mann-Whitney U test, p<1 × 10$^{-100}$). No change in *EGFR* TAD boundary distances

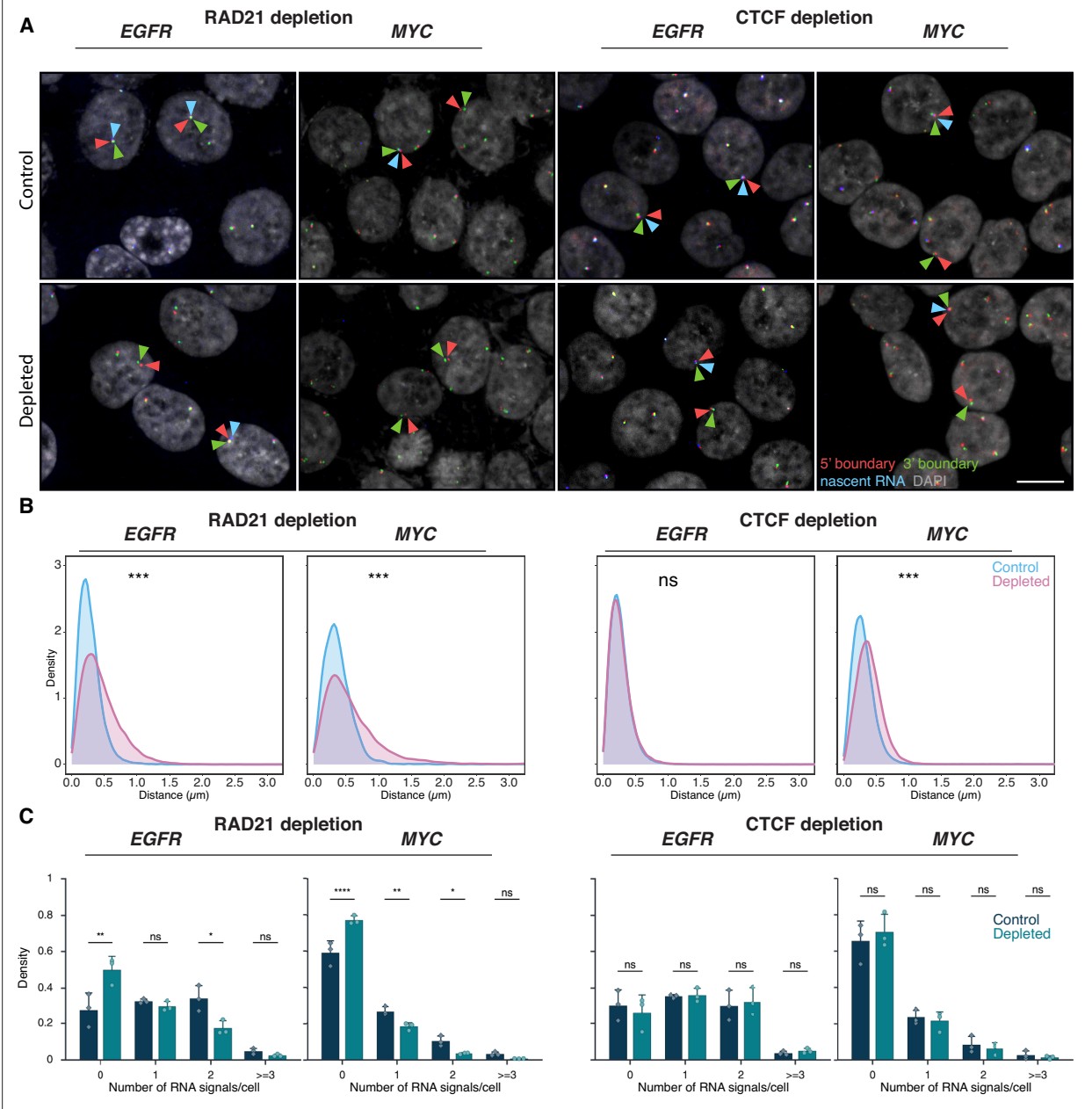

**Figure 6.** Effects of RAD21 and CTCF depletion on topologically associating domain (TAD) boundary distances and gene expression. (**A**) Representative DNA/RNA high-throughput FISH imaging (HiFISH) images of *EGFR* and *MYC* nascent RNA FISH (blue) and its 3' (green) and 5' (red) TAD boundaries DNA FISH in HCT116-RAD21-AID1 and HCT116-CTCF-AID2 cells, respectively, in control and auxin-treated conditions. Scale bar: 10 μm. (**B**) TAD boundary distances and non-TAD controls after RAD21 or CTCF depletion for 3 hr. Values represent an individual dataset from a single experiment representative of multiple experiments. Between 13,000 and 127,500 alleles were analyzed per condition. Mann-Whitney U test p-values are indicated as follows: ***$p<1 \times 10^{-100}$; ns, not significant (p≥0.05). (**C**) Fraction of silent (0), monoallelic (1), biallelic (2), and triallelic or more (≥3) expression of the indicated genes in individual cells after 3 hr or no auxin treatment in HCT116-RAD21-AID1 or HCT116-CTCF-AID2 cells. At least 20,000 cells were measured per experiment. Data represent values from at least two independent experiments (diamonds and circles); diamonds (DMSO control) and circles (RAD21 or CTCF-depleted) represent the mean of means, and error bars indicate SD. p-Values from two-way ANOVA with Bonferroni correction are shown as: ****p<0.0001; **p<0.01; *p<0.05; ns, not significant (p≥0.05).

The online version of this article includes the following figure supplement(s) for figure 6:

**Figure supplement 1.** Depletion of RAD21 and CTCF.

was detected upon CTCF depletion, likely due to the smaller size of the *EGFR* TAD (U test, p≥0.01; *Figure 6C*). Regardless, no significant differences were observed in the expression of *MYC* or *EGFR* following CTCF depletion (Bonferroni-adjusted p-value>0.5; *Figure 6C*). A similar lack of an effect on gene expression was observed for the *ERRFI1* gene upon depletion of CTCF (*Figure 6—figure supplement 1*). Altogether, these results demonstrate that while loss of the cohesin component RAD21 alters boundary distances and reduces gene activity, disruption of TAD boundary architecture by depletion of CTCF does not alter gene expression.

## Discussion

We have used high-throughput DNA/RNA FISH to directly and quantitatively probe the relationship between TAD boundaries and gene activity at the single-cell and -allele level. We find in various experimental settings that TAD boundary proximity is largely unrelated to gene activity.

Uncoupling of TAD boundary structure and gene activity is supported by several observations. TAD boundary distances were indistinguishable between transcriptionally active and inactive alleles across several loci and cell types, even when measured in the same cell nucleus. Furthermore, neither global transcription inhibition nor gene-specific induction of transcription altered TAD boundary proximity, suggesting that short-term transcription dynamics do not affect TAD boundary interactions. These results align with genome-wide findings from population-based studies, indicating that global transcription inhibition does not disrupt TAD structure (*Rao et al., 2017*). A lack of correlation between chromatin domain structure and gene expression has also been noted in early *Drosophila* development, where domain architecture was found to be unrelated to cell-type-specific gene expression patterns (*Espinola et al., 2021*; *Ing-Simmons et al., 2021*), and local chromatin loops formed before the emergence of chromatin domains (*Espinola et al., 2021*; *Ing-Simmons et al., 2021*). In addition, disruption of intra- and inter-TAD interactions in *Drosophila* does not alter the expression of a majority of genes (*Ghavi-Helm et al., 2019*). We also find that while loss of RAD21 altered gene expression, depletion of the architectural boundary protein CTCF did not alter the expression of *MYC* and *EGFR*, nor did it change boundary distances. Combined, these findings suggest that the behavior of TAD boundaries is largely decoupled from gene activity.

These observations point to a model in which the precise demarcation of TAD boundaries plays a relatively minor role in determining the activity of the genes within the TAD. Our results are consistent with the view that transcriptional regulation occurs primarily at finer scales of genome organization—such as enhancer-promoter loops or sub-TAD structures—rather than being governed by the overall configuration of a TAD. In line with this interpretation, we find that depletion of RAD21 reduces *MYC* and *EGFR* expression, likely due to its local effects within the TAD, whereas loss of the bona fide boundary factor CTCF does not alter gene expression. A more local role of chromatin structure on gene expression is also suggested by our finding that the overall TAD structure is not sensitive to the transcriptional status of its genes. Similar observations have been made by others, demonstrating local, intra-TAD effects of transcription on chromatin structure (*Hsieh et al., 2022*; *Luppino et al., 2022*; *Shaban et al., 2024*). The complex interplay of gene expression, local chromatin organization, and TAD structure is further highlighted by the observation of distinct, and only partially overlapping, effects on gene expression upon loss of either of the two cohesin regulators, WAPL and NIPBL (*Schuijers et al., 2018*).

Our findings argue against a role of TADs as stringent regulators of gene expression. One emerging view is that, rather than constituting discrete, stable structures, TADs are probabilistic genome features that represent the integrated sum of all chromatin-chromatin interactions within a genome region (*Natsume et al., 2016*; *Lee et al., 2025*). This interpretation is in line with the observed high degree of single-cell heterogeneity of chromatin interactions, including boundaries (*Finn et al., 2019*), and the documented highly dynamic nature of TAD boundaries which show that the fully formed CTCF loop which defines a specific TAD is a rare event (*Gabriele et al., 2022*). Further support for this view is provided by recent ultra-resolution live-cell imaging, which revealed that at short genomic distances (<200 kb), chromatin loci encounter one another frequently due to spontaneous dynamic motion (*Lee et al., 2025*). Beyond this range, encounter probability declines sharply, and cohesin becomes essential to dynamically bridge distal regulatory elements through active loop extrusion (*Lee et al., 2025*). This extrusion-driven process enables long-range interactions, including between promoters and enhancers and between TAD boundaries (*Lee et al., 2025*). Upon cohesin depletion, these rapid,

distance-independent searches collapse into a diffusive regime reflected as a loss of interactions in both population-based and single-cell analysis (*Natsume et al., 2016*). Our finding that loss of RAD21 has a stronger effect on TAD boundary distance than depletion of the bona fide boundary factor CTCF is consistent with this interpretation. These observations point to a more passive role of TADs in gene regulation, such as limiting inter-TAD enhancer-promoter interactions (*Sood and Misteli, 2022*). In support of a modulatory role rather than a stringent regulatory function, intra-TAD interactions are only enriched ~2-fold compared to inter-TAD interactions (*Finn et al., 2019*). This modulatory behavior does, however, not exclude the possibility of significant effects on gene expression, as has been observed upon deletion of some boundary regions (*Chakraborty et al., 2025*; *Mateo et al., 2019*).

Our study has several limitations. First, our observations are restricted to the relationship of transcription and TAD boundary distances rather than that of structure of the TAD boundary or the TAD as a whole. The behavior of the boundaries may not be representative of the internal TAD architecture. Boundaries may move without affecting the internal compaction or regulatory organization of the domain, and, conversely, the internal structure may change while the boundary distance remains constant. This behavior is in line with the dynamic properties of TADs observed in living cells and highlights the variability of TAD structure at the single-allele level (*Finn et al., 2019*; *Gabriele et al., 2022*). Future work that maps TAD boundaries and internal domain contacts at high resolution in single cells, in parallel with transcriptional state, should provide deeper insight into the relationship between TAD chromatin and transcription. Second, our analysis is limited by the resolution of optical imaging and the size of the FISH probes used. We deliberately use relatively large BAC probes to generate robust, highly reproducible signals and to eliminate effects arising from local chromatin behavior. While the use of larger probes enhances the robustness of measurements, it limits resolution, and subtle changes in boundary architecture may not be detected, although we find very good correlation between Micro-C/Hi-C interaction frequency and distance measurements.

In sum, our observations support the view that the structural and transcriptional layers of genome organization can be partially uncoupled. These insights have implications for interpreting chromatin conformation maps and for understanding the scale at which genome architecture influences transcriptional regulation.

# Materials and methods

## Key resources table

| Reagent type (species) or resource | Designation | Source or reference | Identifiers | Additional information |
|---|---|---|---|---|
| Cell line (*Homo sapiens*) | Cell line HBEC3-KT (HBEC) | *Ramirez et al., 2004* | RRID:CVCL_X491 | Human bronchial epithelial cells immortalized with hTERT and CDK4 |
| Cell line (*Homo sapiens*) | Cell line HFF-hTERT clone 6 (HFFc6; HFF) | Cellosaurus | RRID:CVCL_VC41 | hTERT-immortalized human foreskin fibroblasts (clone 6) |
| Cell line (*Homo sapiens*) | Cell line HCT 116 | Cellosaurus | RRID:CVCL_0291 | Human colorectal carcinoma line; used as parent for degron derivatives |
| Cell line (*Homo sapiens*) | Cell line HCT-116 RAD21-mAID-mClover (RAD21-mAC) | *Natsume et al., 2016* | | Human colorectal carcinoma line (HCT116) derivative used for AID1-mediated RAD21 depletion |
| Cell line (*Homo sapiens*) | Cell line HCT116-CTCF-AID2 | *Yesbolatova et al., 2020* | | HCT116 derivative used for AID2-mediated CTCF depletion |
| Commercial assay or kit | Airway Epithelial Cell Basal Medium | ATCC | ATCC:PCS-300-030 | |
| Commercial assay or kit | Bronchial Epithelial Cell Growth Kit | ATCC | ATCC:PCS-300-040 | |
| Chemical compound, drug | DRB | Sigma-Aldrich | Sigma:D1916 | Transcription inhibition |

*Continued on next page*

*Continued*

| Reagent type (species) or resource | Designation | Source or reference | Identifiers | Additional information |
|---|---|---|---|---|
| Chemical compound, drug | Dexamethasone (Dex) | Sigma-Aldrich | Sigma:D4902 | Glucocorticoid receptor agonist |
| Chemical compound, drug | Auxin | Sigma-Aldrich | Sigma:I3750 | Used for RAD21 depletion in AID1 degron system |
| Chemical compound, drug | 5-Ph-IAA | GLPBio | GLPBio:GC46061 | AID2 ligand used in AID2 degron system |
| Recombinant DNA reagent | BAC probe RP11-112A3 (EGFR upstream control) | BACPAC Resources Center | BACPAC:RP11-112A3 | |
| Recombinant DNA reagent | BAC probe RP11-117I14 (EGFR 5' TAD) | BACPAC Resources Center | BACPAC:RP11-117I14 | |
| Recombinant DNA reagent | BAC probe RP11-98C17 (EGFR 3' TAD) | BACPAC Resources Center | BACPAC:RP11-98C17 | |
| recombinant DNA reagent | BAC probe RP11-788I22 (MYC upstream control) | BACPAC Resources Center | BACPAC:RP11-788I22 | |
| Recombinant DNA reagent | BAC probe RP11-765K23 (MYC 5' TAD) | BACPAC Resources Center | BACPAC:RP11-765K23 | |
| Recombinant DNA reagent | BAC probe RP11-717D13 (MYC 3' TAD) | BACPAC Resources Center | BACPAC:RP11-717D13 | – |
| Recombinant DNA reagent | BAC probe RP11-279H6 (ERRFI1 upstream control) | BACPAC Resources Center | BACPAC:RP11-279H6 | |
| Recombinant DNA reagent | BAC probe RP11-338N10 (ERRFI1 5' TAD) | BACPAC Resources Center | BACPAC:RP11-338N10 | |
| Recombinant DNA reagent | BAC probe RP11-366K21 (ERRFI1 3' TAD) | BACPAC Resources Center | BACPAC:RP11-366K21 | |
| Recombinant DNA reagent | BAC probe RP11-192H11 (VARS2 upstream control) | BACPAC Resources Center | BACPAC:RP11-192H11 | |
| Recombinant DNA reagent | BAC probe RP11-159K11 (VARS2 5' TAD) | BACPAC Resources Center | BACPAC:RP11-159K11 | |
| Recombinant DNA reagent | BAC probe RP11-803D22 (VARS2 3' TAD) | BACPAC Resources Center | BACPAC:RP11-803D22 | |
| Recombinant DNA reagent | BAC probe RP11-107C8 (FKBP5 upstream control) | BACPAC Resources Center | BACPAC:RP11-107C8 | |
| Recombinant DNA reagent | BAC probe RP11-78C20 (FKBP5 5' TAD) | BACPAC Resources Center | BACPAC:RP11-78C20 | |
| Recombinant DNA reagent | BAC probe RP11-828B18 (FKBP5 3' TAD) | BACPAC Resources Center | BACPAC:RP11-828B18 | |
| Sequence-based reagent | Stellaris RNA probe set: EGFR (Atto647N) | LGC Biosearch Technologies | | |
| Sequence-based reagent | Stellaris RNA probe set: MYC (Atto647N) | LGC Biosearch Technologies | | |
| Sequence-based reagent | Stellaris RNA probe set: ERRFI1 (Quasar 670) | LGC Biosearch Technologies | | |
| Sequence-based reagent | Stellaris RNA probe set: VARS2 (Atto647N) | LGC Biosearch Technologies | | |
| Sequence-based reagent | Stellaris RNA probe set: FKBP5 (Atto647N) | LGC Biosearch Technologies | LGC:ISMF-2059-5 | |
| Software, algorithm | HiTIPS | *Keikhosravi et al., 2024*; *Keikhosravi, 2025* | https://github.com/CBIIT/HiTIPS | High-throughput segmentation/detection and quantification pipeline |

*Continued on next page*

*Continued*

| Reagent type (species) or resource | Designation | Source or reference | Identifiers | Additional information |
|---|---|---|---|---|
| Software, algorithm | CellPose | PMID:33318659 | RRID:SCR_021716 | Nucleus segmentation within HiTIPS workflow |
| Software, algorithm | DNA/RNA registration (cross-correlation) | *Keikhosravi, 2024*; *Almansour et al., 2024* | https://github.com/CBIIT/DNA_RNA_registration | Used for sequential DNA/RNA image registration; implemented in Python 3.8 |
| Software, algorithm | Analysis scripts for DNA/RNA HiFISH quantification | Other | https://github.com/CBIIT/mistelilab-tad-ge | R scripts used for boundary-distance and single-cell gene-expression calculations |
| Software, algorithm | UCSC Genome Browser | *Nassar et al., 2023* | RRID:SCR_005780 | |
| Software, algorithm | 4DN Data Portal | *Dekker et al., 2017* | RRID:SCR_016925 | |
| Other | Charcoal-stripped fetal bovine serum | R&D Systems | R&D:S11650H | |
| Other | 384-Well imaging plates (PhenoPlate) | Revvity | Revvity:6057500 | High-throughput imaging format |
| Other | DY549P1-dUTP | Dyomics | | Used for BAC probe labeling by nick translation |
| Other | DY488-dUTP | Dyomics | | Used for BAC probe labeling by nick translation |
| Other | THE RNA Storage Solution | Thermo Fisher Scientific | Thermo Fisher:AM7001 | Included in hybridization buffer |
| Other | Yokogawa CV8000 spinning-disk confocal microscope | Yokogawa | | 60× water objective (NA 1.2); 4 laser lines (405/488/561/640 nm) |

## Cell culture

The identity of cell lines was authenticated by sequencing, and cell lines were tested periodically for absence of mycoplasma. Cell lines used were: HBEC3-KT, HBEC-MYC-MS2, HFF-hTERT clone 6, HCT116, HCT116-RAD21-mAID-mClover, HCT116-CTCF-AID2. Details on cell lines are provided below.

HBECs (HBEC3-KT) are derived from normal human bronchial tissue and immortalized by stable transduction with hTERT and CDK4 (*Ramirez et al., 2004*). HBEC3-KT cells were maintained in keratinocyte serum-free medium (Thermo Fisher Scientific, cat. no. 17005042) supplemented with bovine pituitary extract (Thermo Fisher Scientific, cat. no. 13028014), human growth hormone (Thermo Fisher Scientific, cat. no. 1045013), and 50 U/ml penicillin/streptomycin (Thermo Fisher Scientific, cat. no. 15070063). HBEC-*MYC*-MS2, a derivative of HBEC with 12xMS2 inserted at the 3′ end of *MYC* on both alleles, constitutively expressing a GFP-MS2 coat protein fusion, was maintained under the same conditions.

Human foreskin fibroblasts (HFFs), immortalized with hTERT (*Benanti and Galloway, 2004*), were cultured in DMEM (Thermo Fisher Scientific, cat. no. 10569010) with 10% fetal bovine serum (Thermo Fisher Scientific, cat. no. 10082147) and penicillin/streptomycin. HCT-116 cells (*Natsume et al., 2016*) were cultured in McCoy's 5A medium (Thermo Fisher Scientific, cat. no. 16600082) supplemented with charcoal-stripped fetal bovine serum (*Yesbolatova et al., 2020*) (R&D Systems, cat. no. S11650H) or with 10% FBS, 2 mM L-glutamine, and 100 U/ml penicillin, and 100 µg/ml streptomycin (Thermo Fisher Scientific, cat. no. 15070063). All cells were grown at 37°C in 5% $CO_2$ and split 1:4 twice weekly. Cells were seeded into 384-well imaging plates (PhenoPlate, Revvity, cat. no. 6057500) and allowed to grow to ~80% confluency prior to experiments (*Finn and Misteli, 2021*).

## Cell treatments

For transcription inhibition, HBECs and HFF cells were treated with 100 µM DRB (Sigma-Aldrich, cat. no. D1916) in culture medium for 2 hr, then fixed as described below. For glucocorticoid-mediated transcriptional stimulation, HBECs were cultured in Airway Epithelial Cell Basal Medium (ATCC, cat. no. PCS-300-030) supplemented with the Bronchial Epithelial Cell Growth Kit (ATCC, cat. no. PCS-300-040) and penicillin/streptomycin. To eliminate background glucocorticoid activity, cells were

transferred 24 hr before induction to hormone-free medium, prepared by supplementing the basal ATCC Airway Epithelial Medium with HLL Supplement, L-glutamine, and penicillin/streptomycin while omitting both the P-extract and the Airway Epithelial Cell Supplement, as these components contain glucocorticoids. Dex (Sigma-Aldrich, cat. no. D4902) was prepared as a 100 µM stock solution in ethanol, aliquoted, protected from light, and stored at –20°C. For induction, HBECs were treated with 100 nM Dex (final concentration) for 2 or 4 hr. For RAD21 depletion, HCT116-RAD21-AID1 cells were treated with 0.17 µM auxin (Sigma-Aldrich, cat. no. I3750) or DMSO vehicle control for 3 hr (*Rao et al., 2017*; *Natsume et al., 2016*). Cells were fixed with 4% PFA (Electron Microscopy Sciences, cat. no. 15710) in PBS (Millipore Sigma, cat. no. D8537) for 10 min, washed, and stored in PBS at 4°C. For CTCF depletion, HCT116-CTCF-AID2 cells were treated with 1 µM 5-Ph-IAA (GLPBio, cat. no. GC46061) or DMSO vehicle control. The 5-Ph-IAA (AID2 ligand) working solution was freshly prepared from a 1 mM intermediate stock immediately before use. Cells were cultured for approximately 36 hr to reach 80–90% confluence before treatment. The medium was then replaced with an equal volume of medium containing 1 µM 5-Ph-IAA or an equivalent volume of DMSO control, and cells were incubated for 3 hr at 37°C. Following treatment, cells were directly fixed without washing in 4% paraformaldehyde (Electron Microscopy Sciences, cat. no. 15710) in PBS (Millipore Sigma, cat. no. D8537) for 10 min, washed, and stored in PBS at 4°C until further processing.

## FISH probes

BAC probes targeting *MYC* or *EGFR* TAD boundaries (RP11-765K23, RP11-717D13 for *MYC*; RP11-366D3, RP11-98C17 for *EGFR*) were obtained from the BACPAC Resources Center (BACPAC Resources Center). Negative controls targeting equidistant upstream regions were RP11-788I22 and RP11-112A3 (see *Supplementary file 1* for a complete list of BAC probes). BAC probes were labeled by nick translation at 14°C for 80 min using DY549P1-dUTP or DY488-dUTP (Dyomics) as previously described (*Finn and Misteli, 2021*). Labeled probes were ethanol-precipitated with 38 ng/µl Cot-1 DNA (Millipore Sigma), 256 ng/µl yeast tRNA, and 0.1 M sodium acetate, washed, and resuspended in hybridization buffer, which is made up of 30% formamide (pH 7.0), 10% dextran sulfate, 0.5% Tween-20, 2× SSC, 0.5× RNAsecure RNAse inhibitor, and 3% THE RNA Storage Solution (Thermo Fisher Scientific, cat. no. AM7001) dissolved entirely in molecular $H_2O$. Stellaris RNA probes (LGC Biosearch Technologies) targeting intron 1 of *MYC* and *EGFR* consisted of 48 20-mer oligonucleotides labeled with Atto647N (see *Figure 2—figure supplement 2* for details).

## DNA/RNA HiFISH in 384-well plates

HiFISH was performed as described (*Almansour et al., 2024*). In brief, cells were permeabilized in 0.5% saponin/0.5% Triton X-100/1× RNAsecure in PBS for 20 min, deproteinated in 0.1 N HCl for 15 min, neutralized in 2× SSC, and equilibrated in 50% formamide/2× SSC overnight at 4°C. Hybridization mixtures (0.4 µg DNA probe+12.5 µM RNA probe+hybridization buffer) were denatured at 85°C for 7 min and applied to cells, followed by 48 hr incubation at 37°C. Post-hybridization washes included 50% formamide/2× SSC at 37°C (2×1 hr), 2× SSC (twice), and prewarmed 1× SSC and 0.1× SSC at 45°C (three washes each). Nuclei were stained with 3 µg/ml of 4′,6-diamidino-2-phenylindole (DAPI) in 2× SSC.

## High-throughput image acquisition

Images were acquired using a Yokogawa CV8000 spinning-disk confocal microscope as described in *Almansour et al., 2024*, with a 60× water objective (NA 1.2), four laser lines (405, 488, 561, 640 nm), and appropriate emission filters. Z-stacks spanning 7 µm (1 µm steps) were collected in four channels.

## Image pre-processing

For simultaneous FISH, images were analyzed directly. For sequential FISH, DNA and RNA images were registered as described (*Almansour et al., 2024*) using cross-correlation algorithms implemented in Python 3.8 to align DAPI signals (GitHub: https://github.com/CBIIT/DNA_RNA_registration).

## High-throughput image analysis

HiTIPS software (*Keikhosravi et al., 2024*) was used for segmentation and detection of FISH signals. Nucleus segmentation used CellPose (*Stringer et al., 2021*), and segmentation parameters were

adjusted per plate, with quality-control overlays to confirm accuracy. Data from each well were consolidated into experiment-wide datasets in R (R Core Team, 2024) using tidyverse (*Wickham et al., 2019*) and other packages (*Barrett, 2024*; *Hester, 2023*). For diploid cells, only cells with two DNA signals and ≤2 RNA signals were included for analysis. For analysis of triploid *MYC* HCT116, only cells with two and three *MYC* DNA signals and ≤3 RNA signals were included for analysis. TAD boundary pairs were identified as closest neighbors of a 5′ and 3′ signal. An active allele was defined based on the presence of an RNA signal within 1 µm of either boundary signal, based on the analysis of RNA-DNA distances in pilot experiments, demonstrating that >95% RNA signals were located within 1 µm.

## Micro-C analysis

Published Micro-C XL data from H1-hESC and HFFc6 cells (*Krietenstein et al., 2020*) were visualized using UCSC Genome Browser (*Nassar et al., 2023*) and 4DN Data Portal (*Dekker et al., 2017*; *Reiff et al., 2022*). Heatmaps were displayed with UCSC Track Settings: Display mode Full, Score Maximum Auto-scale, Draw mode triangle, Color HEX (#000000), and no interaction distance filter.

## ChromHMM analysis

Chromatin state annotation for the HCT116 cell line was obtained from ENCODE (file accession ENCF-F993RQV, annotation ID ENCSR448SWW, hg38) using the ChromHMM algorithm (*Ernst and Kellis, 2012*). Each genomic interval was assigned to 1 of 15 functional states (ENCODE Project Consortium, 2020; https://www.encodeproject.org/).

## Data analysis

Distance calculations were performed in R using SpatialTools (*French, 2023*). Statistical comparisons used Kolmogorov-Smirnov (*Massey, 1951*), Wilcoxon rank-sum (*Wilcoxon, 1945*), and Dunn's test with Bonferroni correction (*Dunn, 1964*; *Shaffer, 1995*). p-Values were categorized as ***p<0.001, **p<0.01, *p<0.05, ns p≥0.05.

## Statistical tests

KS tests compared distributions of radial distances; Wilcoxon tests compared medians of two groups; Dunn's test followed Kruskal-Wallis ANOVA for multi-group comparisons (*Kruskal and Wallis, 1952*). Statistical thresholds and exact p-values are reported in figure legends.

## Figure generation

BioRender was used to generate *Figures 1A–3A* (https://biorender.com/vjsbft3).

## Acknowledgements

We thank members of the Misteli lab for input throughout the study. RNA-seq data was kindly provided by Thomas Johnson, NCI. Computation was performed on the NIH HPC Biowulf cluster. FA was supported by a graduate fellowship from the Ministry of Education of Saudi Arabia. This research was supported by the Intramural Research Program of the National Institutes of Health (NIH), National Cancer Institute NCI, Center for Cancer Research through grant 1-ZIA-BC010309 to TM, grant 1-ZIC-BC-011567 to HiTIF, and grant 1-ZIA-BC-011383 to DL. The contributions of the NIH author(s) were made as part of their official duties as NIH federal employees, are in compliance with agency policy requirements, and are considered Works of the United States Government. However, the findings and conclusions presented in this paper are those of the author(s) and do not necessarily reflect the views of the NIH or the U.S. Department of Health and Human Services.

## Additional information

### Funding

| Funder | Grant reference number | Author |
|---|---|---|
| National Institutes of Health | 1-ZIA-BC010309 | Tom Misteli |
| National Institutes of Health | 1-ZIC-BC-011567 | Gianluca Pegoraro |
| National Institutes of Health | 1-ZIA-BC-011383 | Daniel R Larson |
| Ministry of Education of Saudi Arabia | PHD fellowship | Faisal Almansour |

The funders had no role in study design, data collection and interpretation, or the decision to submit the work for publication.

### Author contributions

Faisal Almansour, Conceptualization, Data curation, Formal analysis, Methodology, Writing – original draft, Writing – review and editing; Nadezda A Fursova, Daniel R Larson, Resources, Writing – review and editing; Adib Keikhosravi, Software; Kathleen S Metz Reed, Software, Writing – review and editing; Gianluca Pegoraro, Software, Supervision, Funding acquisition, Writing – review and editing; Tom Misteli, Conceptualization, Formal analysis, Supervision, Funding acquisition, Writing – original draft, Project administration, Writing – review and editing

### Author ORCIDs

Faisal Almansour ⓘD https://orcid.org/0000-0003-0008-9917
Daniel R Larson ⓘD https://orcid.org/0000-0001-9253-3055
Tom Misteli ⓘD https://orcid.org/0000-0003-3530-3020

Reviewer #2 (Public review): https://doi.org/10.7554/eLife.110197.3.sa1
Reviewer #3 (Public review): https://doi.org/10.7554/eLife.110197.3.sa2
Author response https://doi.org/10.7554/eLife.110197.3.sa3

---

## Additional files

### Supplementary files

MDAR checklist

Supplementary file 1. RNA probe sequences.

### Data availability

Code for DNA/RNA image registration is available at https://github.com/CBIIT/DNA_RNA_registration (*Almansour et al., 2024*). The HiTIPS source code is available at https://github.com/CBIIT/HiTIPS, (*Keikhosravi, 2025*) with full documentation-—including package structure, functions, installation instructions, user guidance, output table descriptions, and developer resources-—accessible at https://hitips.readthedocs.io/en/latest/ (*Almansour et al., 2024*). All R scripts used for image quantification, including calculations of TAD boundary distances and single-cell gene-expression measurements from DNA/RNA HiFISH data, are publicly available at https://github.com/CBIIT/mistelilab-tad-ge (copy archived at *Pegoraro, 2026*). Uncropped ImageJ-generated TIFF composites underlying the microscopy panels, along with the DNA/RNA HiFISH datasets used in this study, have been deposited in Figshare (DOI: https://doi.org/10.6084/m9.figshare.31190179, https://doi.org/10.6084/m9.figshare.31399215).

The following datasets were generated:

| Author(s) | Year | Dataset title | Dataset URL | Database and Identifier |
|---|---|---|---|---|
| Almansour F, Fursova NA, Keikhosravi A, Reed KSM, Larson DR, Pegoraro G, Misteli T | 2025 | Confocal Microscopy Raw Images for Figure Generation | https://doi.org/10.6084/m9.figshare.31190179 | figshare, 10.6084/m9.figshare.31190179 |
| Almansour F, Pegoraro G | 2026 | input_figure_1_c | https://doi.org/10.6084/m9.figshare.31399215 | figshare, 10.6084/m9.figshare.31399215 |

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
