## [Editor Report · eLife Assessment]

In this **important** study, DNA and RNA are co-imaged in single cells to show that the proximity of topologically associated domain (TAD) boundaries is uncoupled from the transcriptional activity of nearby genes. The evidence supporting these conclusions is **convincing** for the regions examined, with high-throughput imaging providing robust statistics. This work will be of interest to researchers studying genome architecture and its relationship to gene regulation.

---

## [Referee Report · Reviewer #2 (Public review)]

Summary:

Almansour et al., investigate whether the proximity of TAD boundaries is directly linked to gene activity. The authors use high-throughput imaging to simultaneously measure the gene activity and physical distances between boundary regions in an allele-specific manner. Using transcriptional inhibitors, expression induction, and acute depletion of CTCF and cohesin, they test whether proximity of boundaries affects, or is affected by, gene activity.

Strengths:

The combined use of DNA and RNA imaging enabled simultaneous measurement of boundary proximity and transcriptional status at individual alleles. This allows single-allele correlation between boundary proximity and gene activity at multiple loci across thousands of alleles.

The use of both transcription inhibitors and transcription stimulation provides compelling and consistent evidence that boundary proximity can be disconnected from a gene's activity. The data convincingly support the conclusion that stable proximity between boundary regions is not required for ongoing transcription at the loci and timescales examined.

This work strengthens the emerging view that genome organization at the level of domain boundaries does not impose a deterministic control over transcription.

Strong disruption of boundary distances is only observed upon depletion of cohesin. Notably, this corresponds with the largest changes in gene activity. In contrast, depletion of CTCF actually had minimal impact on boundary distances and also had minimal impact on gene activity. This makes sense in light of previous work, where live cell imaging demonstrated that cohesin is more important for domain-structure, whereas CTCF is only important for blocking cohesin from continuing on, such that the fully formed loop occurs in a very small percentage of cells. Therefore, the fact that disruption of cohesin (more important for internal domain structure) affects gene activity while disruption of CTCF does not is exceptionally interesting.

Weaknesses:

In untreated cells, the distribution of distance measurements between boundary probes is exceptionally narrow. While depletion of RAD21 clearly demonstrates an ability to detect changes in this distribution, this tight baseline distribution may limit sensitivity to more subtle changes (like those one might expect from transcriptional influences).

This approach primarily tests the role of boundary interactions rather than domain organization as a whole.

---

## [Referee Report · Reviewer #3 (Public review)]

Summary:

This study addresses a central question in genome organization: whether the positions of chromosomal domain boundaries are functionally coupled to gene activity. The authors use high-throughput imaging to simultaneously measure distances between boundary markers and nascent RNA production in thousands of individual cells, enabling direct comparison of boundary positions and transcriptional status at single chromosomal copies. This approach is applied across multiple loci, genes, and cell types, and is combined with acute transcriptional perturbations and depletion of architectural proteins to test the relationship between chromosome structure and gene activity in both directions.

This work makes a meaningful contribution by providing direct, single-cell evidence that domain boundary positions and gene activity are largely uncoupled in this system.

Strengths:

A major strength of the work is its single-cell, single-allele resolution, which overcomes the averaging inherent to population-based assays. The authors consistently find that boundary proximity is largely independent of transcriptional status: active and inactive alleles have similar boundary distances, transcriptional perturbations do not shift boundary distributions, and depletion of the boundary factor CTCF does not alter gene expression, whereas cohesin depletion affects both boundary organization and transcription. These conclusions are supported by large numbers of alleles, multiple loci and cell types, and internal controls that distinguish boundary-specific effects from broader chromatin influences. The study offers a robust, scalable imaging pipeline that will be valuable for future studies linking genome organization and transcription at single-cell resolution.

Weaknesses:

The study has important limitations that are acknowledged by the authors. Measurements are restricted to distances between flanking boundaries and do not capture internal domain architecture, sub-domain structure, or finer-scale regulatory contacts. Resolution is limited by probe size and imaging, potentially masking subtle positional changes, and only a small set of loci is examined, leaving open how broadly the uncoupling generalizes. Some perturbation effects, particularly for RAD21, may involve mechanisms beyond boundary disruption.

---

## [Author Response]

The following is the authors’ response to the original reviews.

**Public Reviews:**

**Reviewer #1 (Public review):**
(1) Conceptual framing and interpretation:The central conclusion may require more precise framing to avoid potential overreach. The authors' interpretation equating "physical distance between TAD boundaries" with overall "TAD boundary architecture," and "transcriptional bursting events" with broader "gene activity," could benefit from clarification. This framing may not fully capture the temporal dynamics of transcription or the regulatory complexity within TADs. Furthermore, the broad conclusion of an uncoupled relationship appears to challenge extensive prior evidence from perturbation studies showing that disrupting TAD boundaries can alter gene expression. The authors' own observation of reduced gene activity upon RAD21 degradation suggests that global TAD disruption can affect transcription. A more precise and limited conclusion, acknowledging that their data demonstrate a lack of detectable correlation between boundary distance and bursting activity in their system, would be more accurate and help reconcile these findings with the existing literature.

We have modified statements throughout the manuscript, including in the title, to enhance the precision of our conclusions to avoid overreach. We have also added on p. 16 of our Discussion, a separate section on the limitations of the study, noting that our conclusions are limited to TAD boundary distances and do not reflect the structure of TAD boundaries or of TADs themselves. We have also expanded our Discussion of possible TAD functions on p. 14/15.

(2) Technical methods and data presentation:(2.1) Accuracy and dimensionality of distance measurements: The manuscript does not clearly state whether distances are measured in 2D or 3D, nor does it sufficiently address precision limits. The stated Z-step size (1 µm) may be inadequate for accurately measuring sub-micron chromatin distances in 3D.

We state in both the Results and Methods that our data represent 2D distances derived from maximal-intensity projections of 3D image stacks. We previously published a detailed analysis of the precision of this measurement approach applied to chromatin interactions and documented the effect of 2D vs 3D analysis on these types of measurements. This study by Finn et al., 2022 is cited in the text. We also show in Figure S3 and mention on p. 6 and 10 that we observe similar results using either 2D or 3D analysis.

(2.2) Probe design and systematic error: The genomic coverage size of the BAC probes used for DNA FISH is not explicitly stated. Large probe coverage could inherently blur the precise spatial location of adjacent DNA loci. The reported average distance (~300 nm) may be influenced by the physical size of the probes, as well as systematic expansion or distortion introduced by sample fixation and FISH processing. Although such technical limitations are currently unavoidable, the authors should clarify how these factors might affect their ability to detect subtle distance changes.

The genomic location and size of all probes are provided in Supplementary Table 1. We deliberately use relatively large BAC probes both to generate robust, highly reproducible signals and to eliminate effects arising from local chromatin behavior. In line with earlier characterization of BAC probes (Finn et al., Cell, 2019; Finn et al., Methods, 2022), we find a strong correlation between micro-C/Hi_C interaction frequency and distance measurements. Systematic errors such as sample fixation and FISH processing have previously been evaluated by comparison to live cell data (see Finn et al., 2019) and found to be negligible, especially as all our analyses involve pairwise comparisons, which would both be similarly affected by systematic errors. We discuss resolution limits due to probe size in our new section on study limitations on p. 16.

(2.3) Data Visualization: The manuscript would benefit from including representative, zoomed-in regions of interest from the raw imaging data. This would allow readers to visually assess measured distance differences against background noise.

Raw images for inspection at any magnification are available at here.

(2.4) Potential impact of resolution limits: In Figure 5, the micro-C data reveal a clear difference in interaction patterns inside versus outside the VARS2 locus TAD, yet the imaging data show no corresponding distance difference. This strongly suggests that the current imaging system, limited by optical resolution, probe size, and localisation accuracy, may be unable to resolve finer-scale spatial reorganizations associated with specific chromatin conformations (e.g., enhancer-promoter loops). The authors should explicitly discuss that their conclusion of "no coupling observed" may be constrained by the resolution and sensitivity of their method and does not preclude the possibility of detecting such associations with higher-precision measurements or in live-cell dynamics.

We generally see good agreement between micro-C/Hi-C data and distance measurements. Specifically, we consistently find closer proximity of boundaries than non-boundaries and larger boundary distances for larger TADs than for smaller ones, as presented throughout the study. Contrary to the reviewer’s statement, this is also true for the *VARS2* TAD, where we find statistically significant shorter boundary distances for boundary probes (350 nm) vs the outside control region (390 nm), which correlates with the difference in micro-C interaction score of 5847 vs 2308. These data are shown in Figure 3. Regardless, we mention the issue of resolution due to probe size in the study limitation section on p. 16.

**Reviewer #2 (Public review):**
In untreated cells, the distribution of distance measurements between boundary probes is exceptionally narrow. While depletion of RAD21 clearly demonstrates an ability to detect changes in this distribution, this tight baseline distribution may limit sensitivity to more subtle changes (like those one might expect from transcriptional influences). In addition, the correlation analysis is asymmetric, primarily stratifying by transcriptional status and then comparing boundary distances. Given the central claim that boundary architecture does not influence gene activity, the analysis should be done from the opposite perspective (stratifying by boundary distance).

We mention the limitations on resolution of our approach in our discussion of study limitations on p. 16. An example of an analysis of stratifying by boundary distance is presented in Figure S3C. The conclusion is the same as stratifying by activity status.

Strong disruption of boundary distances is only observed upon depletion of cohesin. Notably, this corresponds with the largest changes in gene activity. In contrast, depletion of CTCF actually had minimal impact on boundary distances and also had minimal impact on gene activity. This makes sense in light of previous work, where live cell imaging demonstrated that cohesin is more important for domain-structure, whereas CTCF is only important for blocking cohesin from continuing on, such that the fully formed loop occurs in a very small percentage of cells. Therefore, the fact that disruption of cohesin (more important for internal domain structure) affects gene activity while disruption of CTCF does not is exceptionally interesting but is lacking from the discussion.

We mention the stronger effect of cohesion depletion compared to CTCF loss on gene expression in multiple locations in the Results and Discussion.

On a related note, this approach primarily tests the role of boundary interactions rather than domain organization as a whole, and it should be acknowledged that internal domain structures are not directly assessed.

We have modified statements throughout the manuscript to clearly indicate that our conclusions relate to boundary interactions rather than domain organization as a whole. We also discuss this in our section on study limitations.

The comparison to work in other organisms (particularly the comparisons made to Drosophila) should be handled with care. The mechanisms underlying domain formation differ substantially across these systems, particularly regarding the differences in CTCF's role.

We have modified our discussion of the data on Drosophila TADs, particularly as it relates to CTCF.

**Recommendations for the authors:**

**Reviewer #1 (Recommendations for the authors):**
I couldn't locate the image data from figshare with the information provided (DOI: 10.6084/m9.figshare.30728354)

The link has been updated

https://figshare.com/projects/_b_TAD_boundaries_and_gene_activity_are_uncoupled_b_/271078.

**Reviewer #2 (Recommendations for the authors):**
Some of the conclusions overreach. I recommend revising the claims and discussion to focus solely on the proximity of boundaries, instead of TADs themselves. This would match better with your experiments.

We have modified statements throughout the manuscript, including in the title, to enhance the precision of our conclusions to avoid overreach. We have also added on p. 16, a separate section on limitations of our study, noting that our conclusions are limited to TAD boundary distances and do not reflect on the structure of the TADs themselves. We have also expanded our Discussion of possible TAD functions on p. 14/15.

I do disagree with the interpretation of the data in some parts, particularly at the end, where you state that disruption of TADs does not impact gene activity. For example, "Altogether, these results demonstrate that disruption of TAD boundary architecture is insufficient to alter gene expression" doesn't seem to match the results. Sure, depletion of CTCF minimally impacted gene expression, but it also minimally impacted the boundary distances. I think it is interesting that depletion of RAD21 had a bigger impact on both gene expression and boundary distances, and this should be discussed.

We have deleted this statement and now mention on p. 13 that RAD21 depletion affected gene expression, whereas loss of CTCF did not, and on p. 15 that loss of RAD21 had a greater impact on boundary distances than loss of CTCF. We have also expanded our Discussion of possible TAD functions on p. 14/15.

Related to this, I also recommend expanding the discussion of prior live-cell imaging work (ref 32) that showed that the fully formed CTCF loop is a rare event.

We have expanded the discussion of prior live-cell imaging work in several locations.

All the analysis is done from the perspective of the gene expression (e.g. group by expression and then measure distances). It would help to show that the inverse analysis is consistent (e.g. group by distances and measure gene expression).

Analysis of data stratified by distance measurements is shown in Figure S3C.

The discussion of the Drosophila work is strange, given that CTCF in Drosophila has a very different N-terminus, explaining why it doesn't really form loops. Sure, maybe it contributes to domains in some way, but probably no more than the dozens of other architectural proteins that have been found in that system. This work clearly focuses on CTCF-loop domains, so I would be specific about that. In the introduction, you do a good job of saying "in human cells, TADs are.... marked by binding sites for the CTCF protein". However, then you overgeneralize and state that TADs form via a process of loop extrusion. I think a simple statement before this to say that TADs in human cells have become somewhat synonymous with CTCF loop domains, and that is how you will use the term here. However, other organisms have TADs despite the lack of conservation of the CTCF protein.

We have modified the text accordingly.

On a related note, in the discussion, you cite two papers in Drosophila to state that "TADs form prior to the establishment of cell-type-specific gene expression programs", but that's not entirely accurate for those papers. They actually show that TADs occur coincident with ZGA, but loops form before that (ref 23: Espinola et al), or that there are indeed a few boundaries that show up before ZGA, but these correspond to RNA Polymerase (ref 24: Ing-Simmons et al.).

We have corrected this statement.